# Mitochondrial Dynamics of Proximal Tubular Epithelial Cells in Nephropathic Cystinosis

**DOI:** 10.3390/ijms21010192

**Published:** 2019-12-26

**Authors:** Domenico De Rasmo, Anna Signorile, Ester De Leo, Elena V. Polishchuk, Anna Ferretta, Roberto Raso, Silvia Russo, Roman Polishchuk, Francesco Emma, Francesco Bellomo

**Affiliations:** 1Institute of Biomembranes, Bioenergetics and Molecular Biotechnology (IBIOM), National Research Council (CNR), 70124 Bari, Italy; a.ferretta@ibiom.cnr.it; 2Department of Basic Medical Sciences, Neurosciences and Sense Organs, University of Bari “Aldo Moro”, 70124 Bari, Italy; anna.signorile@uniba.it (A.S.); silvia.russo92@gmail.com (S.R.); 3Renal Diseases Research Unit, Genetics and Rare Diseases Research Division, Bambino Gesù Children’s Hospital—IRCCS, 00146 Rome, Italy; ester.deleo@opbg.net (E.D.L.); roberto.raso@opbg.net (R.R.); 4Telethon Institute of Genetics and Medicine, 80078 Pozzuoli, Italy; epolish@tigem.it (E.V.P.); polishchuk@tigem.it (R.P.); 5Division of Nephrology, Department of Pediatric Subspecialties, Bambino Gesù Children’s Hospital—IRCCS, 00165 Rome, Italy; francesco.emma@opbg.net

**Keywords:** Fanconi syndrome, nephropathic cystinosis, mitochondrial dynamics, cysteamine, mitochondrial fusion, mitochondrial fission, mitochondrial cristae

## Abstract

Nephropathic cystinosis is a rare lysosomal storage disorder caused by mutations in *CTNS* gene leading to Fanconi syndrome. Independent studies reported defective clearance of damaged mitochondria and mitochondrial fragmentation in cystinosis. Proteins involved in the mitochondrial dynamics and the mitochondrial ultrastructure were analyzed in *CTNS*−/− cells treated with cysteamine, the only drug currently used in the therapy for cystinosis but ineffective to treat Fanconi syndrome. *CTNS*−/− cells showed an overexpression of parkin associated with deregulation of ubiquitination of mitofusin 2 and fission 1 proteins, an altered proteolytic processing of optic atrophy 1 (OPA1), and a decreased OPA1 oligomerization. According to molecular findings, the analysis of electron microscopy images showed a decrease of mitochondrial cristae number and an increase of cristae lumen and cristae junction width. Cysteamine treatment restored the fission 1 ubiquitination, the mitochondrial size, number and lumen of cristae, but had no effect on cristae junction width, making *CTNS*−/− tubular cells more susceptible to apoptotic stimuli.

## 1. Introduction

Nephropathic cystinosis (MIM 219800) is a rare inherited metabolic disease characterized by an impaired transport of the amino acid cystine out of lysosomes due to reduced or absent function of the specific carrier cystinosin, which is encoded by *CTNS* gene [1,2,3]. Kidneys are affected at the initial stage of the disease, leading to early onset Fanconi syndrome, which is characterized by polyuria, glycosuria, phosphaturia, aminoaciduria, and urinary loss of electrolytes and low-molecular-weight proteins [4]. The cystine-depleting agent, cysteamine (MEA), significantly delays symptoms [5,6], but does not treat Fanconi syndrome and is ineffective to prevent the progression of kidney disease. Fanconi syndrome has also been reported in children with specific mitochondrial syndromes [7]. Renal tubular cells are very rich in mitochondria due to the intense reabsorption and excretion processes that occur in this district. We recently reported, in *CTNS*−/− cells derived from proximal tubules, mitochondrial fragmentation associated with respiratory chain dysfunction and low mitochondrial 3′,5′-cyclic adenosine monophosphate (cAMP) levels [8]. Furthermore, enhanced apoptosis [9,10], defect of autophagic flux [11,12], and endo-lysosomal dysfunction [13,14] were observed.

Communication between mitochondria and the endo-lysosomal system is complex. Increasing evidences show close relationship between these two cellular compartments [15,16]. Mitochondria constantly undergo fission and fusion processes to adapt to environmental changes in a continuous and balanced way, in order to maintain morphology and regulate cellular ATP levels. Mitochondrial fission regulates the production of new mitochondria and the segregation of damaged mitochondria. In this process, receptor proteins such as mitochondrial fission factor (Mff), mitochondrial fission 1 protein (Fis1), mitochondrial dynamics protein of 49 kDa (MiD49), and mitochondrial dynamics protein of 51 kDa (MiD51) recruit the large GTPase dynamin-related protein 1 (Drp1) from the cytosol to the outer mitochondrial membrane (OMM), which forms a multimeric complex with mitochondrial membrane adaptors [17,18]. Mitochondrial fusion is mediated by three key regulatory fusion proteins: the dynamin-related GTPases mitofusin 1 (MFN1) and mitofusin 2 (MFN2), responsible for the fusion of OMM and the dynamin-related GTPases optic atrophy 1 (OPA1), which mediates fusion of the inner mitochondrial membrane (IMM) and contributes to the maintenance of mitochondrial potential, to control respiratory chain activity and apoptosis [19,20]. When mitochondrial dynamics are impaired, dysfunctional mitochondria are selectively eliminated through mitophagy, which is initiated by ubiquitin-dependent or ubiquitin-independent signals [21]. The lysosomal alterations in cystinosis lead to defective autophagic clearance of damaged mitochondria [12], therefore the purpose of our study was to investigate mitochondrial dynamics in *CTNS*−/− conditionally immortalized proximal tubular epithelial cells (ciPTEC) carrying the classical homozygous 57-kb deletion in the intent of identifying new therapeutic targets and biomarkers for treatment follow-up.

## 2. Results

### 2.1. CTNS−/− ciPTEC Showed Deregulation of Proteins Involved in Mitochondrial Fission/Fusion Processes

Recruitment of the GTPase dynamin-related protein 1 (Drp1) to mitochondria is a key step required for mitochondrial fission and its reversible phosphorylation was implicated in the regulation of this process. The analysis of phosphorylated Drp1 at Ser-637 (Drp1^pS637^) showed high variability of protein phosphorylation with no significant differences in wild type and CTNS−/− ciPTEC, also 24 h treatment with 100 µM cysteamine (MEA) did not affect Drp1^pS637^phosphorylation (Figure 1). The western blotting of the pro-fission mitochondrial protein Fis1 revealed the ubiquitinated and not-ubiquitinated forms of protein. *CTNS*−/− ciPTEC showed increased level of Fis1 (2.47 ± 0.27 vs. 1.08 ± 0.08, *p* = 0.0027). Treatment with 100 µM MEA for 24 h further increased total Fis1 protein level (2.47 ± 0.27 vs. 3.59 ± 0.15, *p* = 0.011) but almost completely reduced the ubiquitinated counterpart by 96.3% (*p* < 0.001) (Figure 1). Third key regulatory protein analyzed was mitochondrial fission factor (Mff), which localizes on OMM and promotes the recruitment of DRP1 to the mitochondrial surface. This protein, shown in its four isoforms, was not modified in *CTNS*−/− ciPTEC and the expression was unchanged after MEA treatment (Figure 1).

The inner mitochondrial membrane GTPase OPA1 undergoes constitutive processing leading to the conversion of the un-cleaved long OPA1 (L-OPA1) in cleaved short variants (S-OPA1). Various stress conditions, including apoptotic stimuli, trigger the complete conversion of L-OPA1 into S-OPA1. In this regard, *CTNS*−/− ciPTEC were characterized by a significant increase of short variants (52.4%, *p* < 0.05), but 24 h treatment with 100 µM MEA did not show significant effects (Figure 2). In agreement with higher S-OPA1 levels, we found that the active form of mitochondrial metallo-endopetidase OMA1, which catalyze conversion of OPA1 into short isoforms and triggers mitochondrial fragmentation, was increased by 79.8% in *CTNS*−/− ciPTEC (*p* < 0.001), and not rescued by MEA treatment (Figure 2). OPA1 can oligomerize at the inner mitochondrial membrane to keep the cristae junction tight, therefore cell fresh pellets were treated with the cross-linker bis-maleimidohexane (BMH) 1 mM or with vehicle to test the oligomeric state of OPA1. The OPA1 oligomer, immune-revealed as a high molecular-weight band (≈250 kDa), decreased in *CTNS*−/− cells by 23.5% compared to *CTNS*+/+ cells and was not affected by MEA treatment. The absence of OPA1 oligomerization in cells treated with vehicle (DMSO) confirmed the specificity of cross-linking (Figure 2).

The expression of MFN2, an outer mitochondrial membrane GTPase involved in fusion processes, was not changed in *CTNS*−/− ciPTEC with respect to control cells (Figure 3). However, the higher molecular weight band, corresponding to ubiquitinated MFN2, indicated an increase of ubiquitination in *CTNS*−/− ciPTEC by 70.8% (*p* < 0.001). Treatment with MEA showed 37.8% reduction of ubiquitination but the effect was not statistically significant (Figure 3).

Numerous mitochondrial outer membrane proteins are modified with K48- and K63-linked ubiquitin chains, including the mitochondrial fusion factors MFN1 and MFN2 and fission factors Fis1 and Drp1, triggering a cascade of events that result in mitophagy. According with previous results, we found in *CTNS*−/− ciPTEC a significant increase in protein levels of the E3 ubiquitin ligase parkin (2.92 ± 0.22 in *CTNS*−/− vs. 1.0 ± 0.04 in *CTNS*+/+, *p* = 0.001). MEA treatment did not change parkin expression (Figure 4A). Ubiquitin carboxyl-terminal hydrolase 30 (USP30) mediates the removal of the ubiquitin chains added by parkin to ubiquitilated forms of mitofusins, such as MFN2, therefore we analyzed the expression of this deubiquitinating enzyme tethered to the OMM and showed in *CTNS*−/− ciPTEC 62.9% decreasing compared to wild type cells (*p* < 0.001). Treatment with MEA rescued USP30 expression in *CTNS*−/− ciPTEC by 87.9% (*p* = 0.037) (Figure 4B).

### 2.2. Mitochondrial Cristae Organization Was Impaired in CTNS−/− ciPTEC

Comparative ultrastructural analysis of mitochondria in ciPTEC, by transmission electron microscopy (TEM), showed the presence of smaller mitochondria in *CTNS*−/− ciPTEC compared to *CTNS*+/+ ciPTEC (0.12 ± 0.01 µm^2^ in *CTNS*−/− vs. 0.26 ± 0.01 µm^2^ in *CTNS*+/+, *p* < 0.0001). Mitochondrial area was completely recovered by MEA treatment (0.32 ± 0.04 µm^2^ in *CTNS*−/− + MEA vs. 0.12 ± 0.01 µm^2^ in *CTNS*−/−, *p* < 0.0001). Moreover, TEM showed a substantial reduction in number of cristae per mitochondrial section in *CTNS*−/− (5.1 ± 0.6 in *CTNS*+/+ vs. 1.6 ± 0.2 in *CTNS*−/−, *p* < 0.0001). This parameter was rescued by MEA by 167.5% (*p* < 0.0001). Because evidences underlined a critical role of cristae junction in mitochondrial function and organization, we measured them. It was found an increase in cristae junction width (39.65 ± 4.83 nm in *CTNS*+/+ vs. 53.21 ± 10.05 nm in *CTNS*−/−, *p* < 0.0001) and cristae lumen width (29.68 ± 3.53 nm in *CTNS*+/+ vs. 37.04 ± 6.13 nm in *CTNS*−/−, *p* < 0.0003) partially rescued by cysteamine (Figure 5).

## 3. Discussion

Nephropathic cystinosis is a rare inherited metabolic disorder, belonging to the group of lysosomal storage diseases (LSD). The disease is the first cause of Fanconi syndrome in children, characterized by loss of electrolytes, glucose, amino acid, low-molecular weight proteins in urine caused by proximal tubule dysfunction [4,22]. The molecular mechanism at the basis of Fanconi syndrome in cystinosis is not completely understood. Several mechanisms have been suggested to contribute to the pathogenesis of cystinosis, including lysosomal overload, endo-lysosomal transport defect, altered chaperone-mediated autophagy, mTOR signaling, transcription factor EB (TFEB) expression [11,13,23,24,25,26,27]. Cysteamine, a cystine-depleting agent, which allows clearance of cystine from lysosomes, represents the only specific treatment for cystinosis. However, cysteamine does not correct the above cited cellular alterations and does not stop the progress of the Fanconi syndrome. Our recent studies have shown in *CTNS*−/− ciPTEC a higher mitochondrial fragmentation index associated with lower mitochondrial potential and mitochondrial cyclic AMP levels, rescued by 24 h treatment with 100 µM cysteamine or with the cell-permeant analogue of cyclic AMP, 8-Br-cAMP [8]. cAMP, in fact, is one of the major regulators of mitochondrial function [28,29,30] and dynamics [31]. In this contest, it should be noted that MEA has been found to improve mitochondrial function in mitochondrial respiratory chain diseases [32]. Mitochondrial dynamics is balanced between rates of fusion and fission that respond to pathophysiologic signals. This finely regulated equilibrium is closely related to the quality control system, which is mainly ascribed to the ubiquitin protease system (UPS) and to the intra-mitochondrial proteolytic systems [33]. In our experimental model, no significant differences were observed on protein levels of phosphorylated Drp1 at Ser-637. The PKA-dependent phosphorylation of Drp1 at Ser-637 is generally recognized to block Drp1 GTPase activity and to suppress mitochondrial fission. However, and in agreement with data previously reported by Yu et al., Drp1^pS637^ did not contribute substantially to mitochondrial fission regulation in *CTNS*−/− ciPTEC [18]. Several key effector proteins of mitochondrial fusion (MFN1 and MFN2) and fission (Fis1, Mff) are located at the OMM with their domains exposed at the cytosolic side of the membrane. This peculiar topology allows selective removing of fusion or fission components exposed by the UPS, providing fine tuning of this high-level regulatory processes. In mammalian cells, Fis1 accumulates in the mitochondrial outer membrane during the fission process, whereas MFN1 and MFN2 are ubiquitinated and degraded by the proteasome [34]. In this respect, we observed an increase in Fis1, ubiquitinated MFN2 and of the E3 ubiquitin ligase Parkin in *CTNS*−/− ciPTEC, indicating the tendency to mitochondrial fragmentation. The effect of MEA on the reduction of ubiquitilated MFN2 could be ascribed to the modulation of USP30, a mitochondrion-localized deubiquitilase, which counteracts Parkin by deubiquitilating OMM proteins and regulate mitophagy [35]. These findings are consistent with data showing that an increase of parkin expression results in mitochondrial fragmentation [36] and is associated with MFN2 ubiquitination [37]. In addition, the increase of FIS1, in cystinotic cell, might be due to Sirt3 protein [38] that was found downregulated in the same cystinotic cell line [8]. OPA1, an inner mitochondrial membrane GTPase protein, has gained attention because it regulates important mitochondrial functions, including the balance between mitochondrial fusion and fission processes, the stability of the mitochondrial respiratory chain complexes, the proapoptotic release of cytochrome *c* molecules sequestered within the mitochondrial cristae and the maintenance of mitochondrial cristae architecture [39]. The protein expression levels of OPA1 were not significantly changed in mutated cells, compared to control ciPTEC cells (data not shown). However, the activity of OPA1 is also controlled at the post-translational level by proteolytic and acetylation changes [31]. Various stress conditions, including apoptotic stimulation, trigger the complete conversion of L-OPA1 into S-OPA1. The pro-fusion activity of OPA1 depends on the balanced formation of L-OPA1 and S-OPA1 [40]. In this respect, we observed an alteration of OPA1 processing in *CTNS*−/− ciPTEC cells. In particular, cystinotic cells were characterized by a significant increase of S-OPA1, associated with an increase in the protease OMA1 activity, that was not prevented by MEA. In addition to its role as a fusion protein, OPA1 controls the remodeling of mitochondrial cristae. Specifically, OPA1 forms oligomers in the inner mitochondrial membrane that keep the cristae junctions tight. During apoptosis, oligomers are destabilized causing the opening of cristae and release of cytochrome *c* out of the mitochondria. OPA1 oligomers were decreased in cystinotic cells. MEA did not rescue this phenotype. This defect correlates with increased cristae junction width that we observed in our TEM ultrastructural analyses.

In summary, our study shows deregulation of several proteins involved in mitochondrial dynamics in *CTNS*−/− cells. We observed mitochondrial fragmentation in cystinotic cells associated with altered proteolytic processing of OPA1, increased Fis1 and parkin protein levels. The deregulation of parkin could result in increase of ubiquitination of MFN2. The cristae number was decreased while the cristae lumen was increased in cystinotic cells, which parallels the previously reported bioenergetic defects in these cells. The cristae junction width was increased in *CTNS*−/− cells, which is most likely secondary to low OPA1 oligomerization levels. MEA treatment restored mitochondrial size, cristae number, and lumen, but had no effect on cristae junction width, making tubular cells more susceptible to apoptotic stimuli. In this contest, we highlight several cellular mediators of mitochondrial dynamics that could be useful to develop new therapeutic interventions [41].

## 4. Materials and Methods

### 4.1. Cell Culture

Conditionally immortalized proximal tubular epithelial cells (ciPTEC), from healthy donor and cystinotic patients were obtained from Radboud University Medical Center, Nijmegen, The Netherlands and cultured as described in [42]. Cells were grown in a humidified atmosphere with 5% CO_2_ at 37 °C. Where indicated, the cells were treated with 100 μM MEA or water (vehicle) for 24 h.

### 4.2. SDS-PAGE and Western Blotting

Monolayer cell cultures were harvested with 0.05% trypsin, 0.02% EDTA. After trypsinization, cells were centrifuged at 500× *g* and resuspended in RIPA buffer (150 mM NaCl, 5 mM EDTA, 50 mM Tris/HCl, 0.1% SDS, 1% Triton X-100, pH 7.4), in the presence of a protease inhibitor (0.25 mM PMSF). Cell lysate proteins were subjected to SDS-polyacrylamide gel electrophoresis (PAGE), transferred to a nitrocellulose membrane and immunoblotted with antibodies against OPA1 (Thermo scientific, Waltham, MA, USA; Pierce Antibodies); Fis1, Mfn2 (Merck Millipore, Burlington, MA, USA); parkin, USP30 (Santa Cruz Biotechnology, Dallas, TX, USA); OMA1 (Abcam, Cambridge, UK); Drp1, Mff (Cell Signaling, Danvers, MA, USA) and actin (Merck, Kenilworth, NJ, USA). After being washed in TTBS, the membranes were incubated for 1 h with anti-mouse or anti-rabbit IgG peroxidase-conjugate antibody. Immunodetection was performed with the enhanced chemiluminescence (ECL) (Thermo scientific, Waltham, MA, USA). VersaDoc imaging system (BioRad, Milan, Italy) was used for densitometric analysis.

### 4.3. Analysis of OPA1 Oligomers

To investigate on OPA1 oligomerization, cell fresh pellets were treated with the cross-linker bismaleimidohexane (BMH) 1 mM or with vehicle (DMSO) for 30 min at 37 °C. After incubation, the samples were centrifuged, resuspended in SDS lysis buffer and then subjected to SDS-PAGE and western blotting analysis with the antibody against OPA1.

### 4.4. Electron Microscopy

For routine EM the cells were grown in 12 well plates as a monolayer. At the end of the experiment the cells were fixed with 1% Glutaraldehyde prepared in 0.2 M Hepes buffer. Then the cells were scraped, pelleted, post-fixed in OsO4 and uranyl acetate and embedded in Epon as described previously [43]. From each sample, thin 60 nm sections were cut using Leica EM UC7 (Leica Microsystems, Vienna, Austria). EM images were acquired from thin sections using a FEI Tecnai-12 electron microscope (FEI, Eindhoven, Netherlands) equipped with a VELETTA CCD digital camera (Soft Imaging Systems GmbH, Munster, Germany).

## Figures and Tables

**Figure 1 ijms-21-00192-f001:**
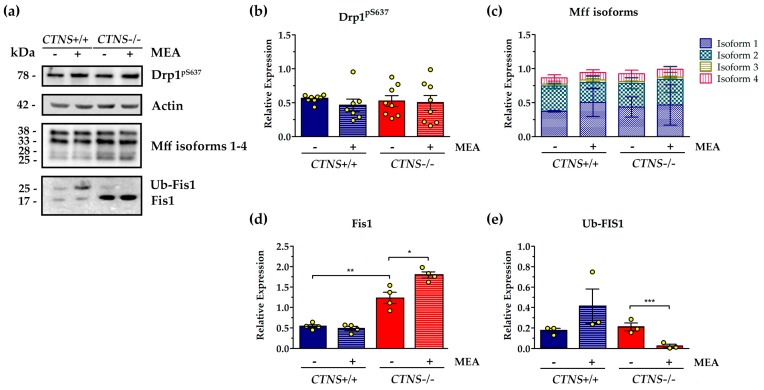
Analysis of proteins involved in fission process of mitochondrial dynamics in untreated and cysteamine (MEA)-treated *CTNS*−/−compared to *CTNS*+/+. Cell cultures were treated with 100 μM cysteamine (MEA) or DMSO (vehicle) for 24 h as specified in the figure. (**a**) Representative immunoblotting analysis in cellular lysate of conditionally immortalized proximal tubular epithelial cells (ciPTEC) from a healthy subject (*CTNS*+/+) and cystinotic patient (*CTNS*−/−). (**b**–**e**) The histograms (Drp1^pS637^, panel (**b**), *n* = 8; mitochondrial fission factor (Mff), panel (**c**), *n* = 3; mitochondrial fission 1 protein (Fis1), panel (**d**), *n* = 4; ubiquitinated Fis1 (Ub-Fis1), panel (**e**), *n* = 3) represent the means values ± SEM of the relative expression normalized on actin level. Densitometric analysis was performed by Versa-Doc imaging system BioRad, using Quantity One software. *p*-value less than 0.05 was considered as statistically significant, (Student’s *t* test, *** *p* < 0.001; ** *p* < 0.01; * *p* < 0.05). For further details see under “materials and methods” section.

**Figure 2 ijms-21-00192-f002:**
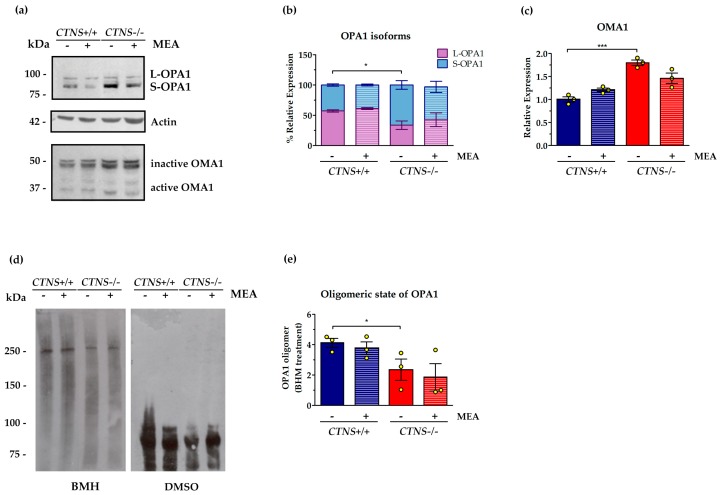
Processing and oligomerization of optic atrophy 1 (OPA1) fusion protein in untreated and MEA-treated *CTNS*−/− compared to *CTNS*+/+. (**a**) Representative immunoblotting analysis of ciPTEC obtained from *CTNS*+*/*+ and *CTNS*−*/*−. Where indicated, the cells were treated with MEA or DMSO (vehicle) for 24 h. The histograms of OPA1 (**b**) represent the percentage of relative expression of L and S forms of OPA1 in each lane (*n* = 3). The histograms of OMA1 (**c**) represent the means values ± SEM of the relative expression normalized on actin level (*n* = 3). (**d**) The fresh collected cells were treated with the cross-linker 1,6-bismaleimidohexane (BMH) 1 mM or with vehicle (DMSO) for 30 min at 37 °C, then centrifuged and resuspended in sodium dodecyl sulfate (SDS) lysis buffer for western blotting analysis with the antibody against OPA1. (**e**) The histograms represent the means values ± SEM of the relative expression of OPA1 oligomers (*n* = 3). Densitometric analysis was performed by Versa-Doc imaging system BioRad, using Quantity One software. Student’s *t* test, *** *p* < 0.001; * *p* < 0.05. For further details see under “materials and methods” section.

**Figure 3 ijms-21-00192-f003:**
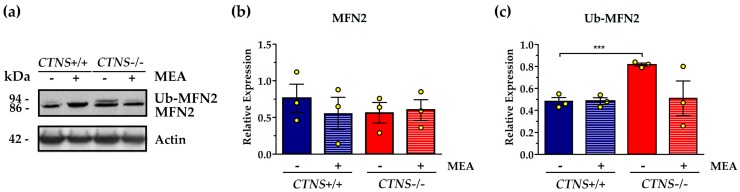
Expression and ubiquitination of mitofusin 2 (MFN2) in untreated and MEA-treated *CTNS*+/+ and *CTNS*−/−. (**a**) Representative immunoblotting analysis of untreated and MEA-treated ciPTEC *CTNS*+/+ and *CTNS*−/−. (**b**) The histogram of MFN2, *n* = 3, and (**c**) the histogram of ubiquitinated MFN2 Ub-MFN2, *n* = 3, represent the mean values ± SEM of the relative expression normalized on actin level. Densitometric analysis was performed by Versa-Doc imaging system BioRad, using Quantity One software. Student’s *t* test, *** *p* < 0.001.

**Figure 4 ijms-21-00192-f004:**
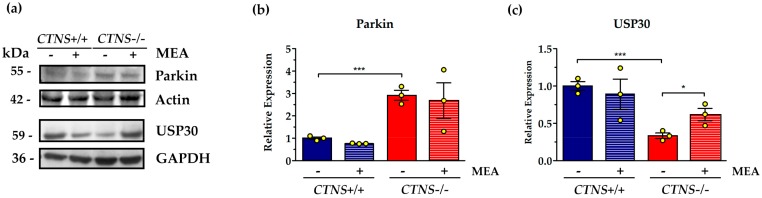
Parkin and ubiquitin carboxyl-terminal hydrolase 30 (USP30) proteins levels and MEA effect in *CTNS*+/+ and *CTNS*−/−. (**a**) Immunoblotting analysis of untreated and MEA-treated ciPTEC *CTNS*+*/*+ and *CTNS*−*/*−. (**b**) The histogram represents the means values ± SEM of the relative expression of Parkin normalized on actin level (*n* = 3). (**c**) The histogram represents the means values ± SEM of the relative expression of USP30 normalized on actin level (*n* = 3). Densitometric analysis was performed by Versa-Doc imaging system BioRad, using Quantity One software. Student’s *t* test, *** *p* < 0.001; * *p* < 0.05.

**Figure 5 ijms-21-00192-f005:**
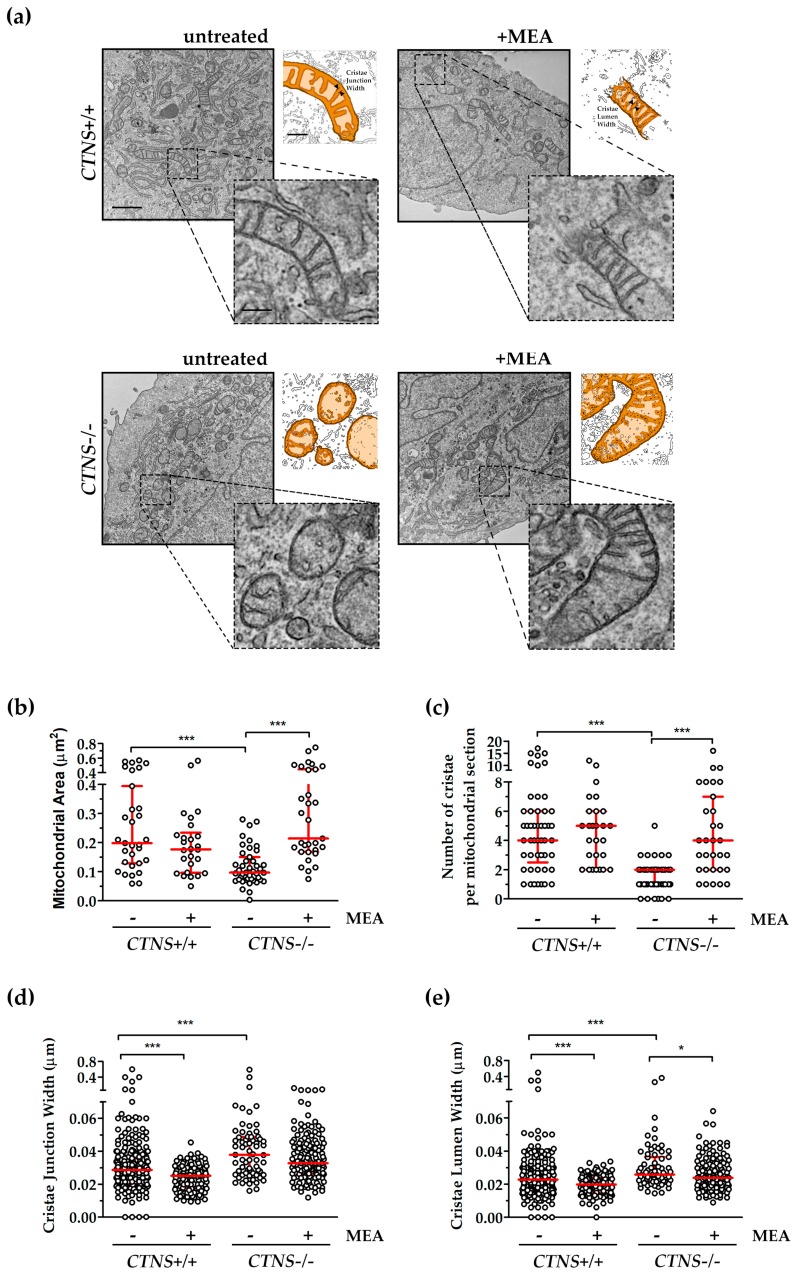
Comparative ultrastructural analysis of mitochondria in ciPTEC. (**a**) Representative images of TEM with magnification 16,000× of untreated and MEA-treated ciPTEC *CTNS*+/+ and *CTNS*−/−; scale bar = 1 µm. As shown in high magnification cropped micrographs and in ad hoc schematic reconstruction, mitochondria kept preserved ultrastructure in ciPTEC *CTNS*+/+ and in MEA-treated ciPTEC *CTNS*+/+ and *CTNS*−/−, whereas ciPTEC *CTNS*−/− showed disruption of mitochondrial cristae and the disarrangement of the internal structures; scale bar = 200 nm. (**b**–**e**) Quantitative analysis was performed with ImageJ v.1.52p in *n* ≥ 5 double-blind acquisitions for each experimental condition, red lines represent median with interquartile range. (**b**) Evaluation of relative mitochondrial size measured as area of *n* ≥ 27 mitochondrial sections. (**c**) Average number of cristae per mitochondrion in each cell (*n* ≥ 27 mitochondrial sections). (**d**) The measure of distance of cristae junction near the inner membrane boundary and (**e**) the measure of cristae lumen, assessed on cristae membranes that outline the lumen boundary, were assessed in *n* ≥ 100 cristae. Non-parametric Mann-Whitney test was applied, *** *p* < 0.001; * *p* < 0.05.

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
