# Peer review of "Mitochondrial Dynamics of Proximal Tubular Epithelial Cells in Nephropathic Cystinosis"

_ijms, 2019, doi:10.3390/ijms21010192_

Round 1

Reviewer 1 Report

In their manuscript De Rasmo and colleagues investigate the relationship between the defective autophagic clearance of damaged mitochondria due to lysosomal dysfunction in nephropathic cystinosis and the mechanisms underlying mitochondrial dynamics in CTNS-/- proximal tubular epithelial cells in order to generate data suitable to be used to identify new therapeutic targets and related biomarkers useful to evaluate treatment efficacy.

The manuscript is potentially very interesting although the study itself seems to be based on a quite reductive approach. Although the manuscript is well written and it is clear that the authors have a good knowledge of the subject they base their conclusions on some western blot analyses and a TEM analysis which provide a too poor set of data to allow to draw conclusions and to be considered as a stand alone study.

Here below few minor comments to be addressed:

The resolution and quality of Figure 1 are very bad: all the texts displayed, including numbers, are unreadable. It seems to have been cut and past as image from an inadequate source. Please fix it.

In the caption: the meaning of the acronym ciPTEC should be reported;  the sentence “After incubation, proteins of cellular lysate were loaded on 8% SDS-PAGE, transferred to nitrocellulose membranes, and immunoblotted with the antibodies described in the figure” should be removed being completely redundant (it reports a protocol routinely performed); the number of experiments for each figure (from “a” to “e”) should be reported (n = ?).

Concerning Figures 2, 3 and 4 please consider the same directions reported for Figure 1 both for the figures and the captions.

In Figure 5 the panels b-e are inadequate and very difficult for the readers to understand. I would suggest to modify the histogram format beyond the format of the text and the resolution which also has to be improved. Panel a is unclear and the related caption should briefly report and explain the message for the reader.

Several careless mistakes and linguistic inaccuracies are found throughout the manuscript including title.

Reviewer 2 Report

This work documents increased parkin levels in immortalized proximal tubular epithelial cells from cystinotic patients relative to normal controls. This results in the ubiquitination of MFN2 and Fis1 leading to mitochondrial fragmentation and increased susceptibility of cystinotic cells to apoptosis. The experiments are well-conceived and appropriately controlled. The only detractor from the impact of this paper is the relatively poor quality of the electron micrographs in the 4 left panels in Figure 5. Even with a magnifying lens, this reviewer was unable to confirm the ultrastructural details referred to in the text and the legend to Figure 5. Since this information is critical to the overall message provided in this manuscript, it is important that higher quality (and magnification) images be substituted in a revised paper.  

Round 2

Reviewer 1 Report

The manuscript has been properly improved and is now suitable for publication.

Reviewer 2 Report

This manuscript has been nicely revised in accord with my previous suggestions. Acceptance is recommended.